# Delamination Detection and Localization in Vibrating Composite Plates and Shells Using the Inverse Finite Element Method

**DOI:** 10.3390/s23187926

**Published:** 2023-09-15

**Authors:** Faraz Ganjdoust, Adnan Kefal, Alexander Tessler

**Affiliations:** 1Faculty of Engineering and Natural Sciences, Sabanci University, Tuzla, Istanbul 34956, Turkey; faraz@sabanciuniv.edu; 2Integrated Manufacturing Technologies Research and Application Center, Sabanci University, Tuzla, Istanbul 34956, Turkey; 3Composite Technologies Center of Excellence, Istanbul Technology Development Zone, Sabanci University-Kordsa Global, Pendik, Istanbul 34906, Turkey; 4Structural Mechanics and Concepts Branch, NASA Langley Research Center, Mail Stop 190, Hampton, VA 23681-2199, USA; tessler.research@gmail.com

**Keywords:** delamination damage, vibrations, laminated composite shells, inverse finite element method, refined zigzag theory

## Abstract

Delamination damage is one of the most critical damage modes of composite materials. It takes place through the thickness of the laminated composites and does not show subtle surface effects. In the present study, a delamination detection approach based on equivalent von Mises strains is demonstrated for vibrating laminated (i.e., unidirectional fabric) composite plates. In this context, the governing relations of the inverse finite element method were recast according to the refined zigzag theory. Using the in situ strain measurements obtained from the surface and through the thickness of the composite shell, the inverse analysis was performed, and the strain field of the composite shell was reconstructed. The implementation of the proposed methodology is demonstrated for two numerical case studies associated with the harmonic and random vibrations of composite shells. The findings of this study show that the present damage detection method is capable of real-time monitoring of damage and providing information about the exact location, shape, and extent of the delamination damage in the vibrating composite plate. Finally, the robustness of the proposed method in response to resonance and extreme load variations is shown.

## 1. Introduction

Industry is becoming dependent on composite materials at an ever-increasing rate. These materials offer superior properties, and consequently are the material of choice for applications in advanced engineering structures readily available in aerospace and marine industries. Nevertheless, composite materials are prone to damage, and the damage mechanisms in composites are more complex than conventional alternatives, such as metals. Among the various damage modes of composite materials, delamination damage is a prevalent and critical failure mode. The composite structures lose their structural integrity because of degradation of stiffness triggered by delamination damage. Delamination can happen in composites during manufacturing processes, or as a result of in-service accidents [1]. What makes the delamination damage detection critical and challenging is the fact that it occurs through the thickness of the laminated material, and it often does not have any subtle surface effects. Additionally, it is worth mentioning that the stacking sequence of the laminate is one of the factors that has control over propagation or prevention of delamination damage [2].

Given the vitality of the issue, important research efforts have been dedicated to devising methodologies and designing experiments for detecting and/or preventing delamination damage in composite materials. Delamination detection via ultrasonic techniques [3], fiber optic sensors [4], acoustic emission [5,6], radiography [7,8,9], thermography [10,11], digital image correlation [12,13], and guided waves [14] are among the most commonly used techniques in the literature. Drilling of composites is one of the most common factors that triggers delamination damage in such materials. In this context, numerous studies in the literature focus on this phenomenon. In terms of experimental efforts to reduce and prevent delamination damage caused by drilling, Capello [15] demonstrated that using supports for the workpiece during drilling decreases drilling-induced delamination drastically. Ultrasonic-assisted drilling was presented as another preventive measure in the study by Mehbudi et al. [16]. Seif et al. [17] introduced a non-destructive monitoring technique based on imaging to measure the size of delamination damage in composite materials. Moreover, De Albuquerque et al. [18] used a data-driven method based on artificial neural networks to select suitable drilling geometries to minimize delamination damage. Jia et al. [19] incorporated thermal effects in their predictive analytical model of drilling thrust force in an attempt to reduce delamination in composite materials. In addition to drilling, impact is one of the causes of delamination damage, and given its minimal surface effects, delamination is classified as a type of barely visible impact damage (BVID) [20]. Johnson et al. [21] developed a predictive continuum model to identify delamination damage caused by impact. Other attempts at detection of impact-induced delamination damage include the application of nonlinear acoustics for identifying delamination [22].

Delamination detection through the mentioned techniques provides promising results; however, most of the efforts in the direction of delamination damage detection in laminated composite materials fall into the category of vibrational structural health monitoring (VSHM). In the work of Ratcliffe and Bagaria [23], fundamental vibrational mode of a composite beam was used to define a damage index, which facilitated locating delamination damage in the plate. This study inspired a study by Barman et al. [24], where they used the bending modes of the composite beam and defined damage indices based on [23]. Next, by solving a minimization problem based on the vibrational characteristics of the beam, the exact location of the damage was detected. Moreover, Kessler et al. [25] proposed a model-based method using the vibrational frequency responses of composite structures. Zhang et al. [26] compared three inverse vibration-based damage detection techniques implemented in beam problems, and highlighted the challenges associated with determining the location of delamination damage. Additionally, data-driven delamination detection approaches were developed within the context of vibration-based monitoring systems [27]. Generally, VSHM methods have proven to be very efficient in terms of identifying the existence of damage in a composite material; however, they fail to provide an accurate location or shape of the delaminated region.

However, development of a shape-sensing approach based on the minimization of a least-squares functional, known as the inverse finite element method (iFEM) [28], provided researchers with the means to reconstruct the displacement and strain fields of engineering structures, irrespective of their material models and/or their loading conditions. Gherlone et al. [29] have shown that the iFEM is more versatile than other available shape-sensing techniques, such as Ko’s displacement method [30], modal methods [31], or shape-sensing based on artificial intelligence [32], in terms of implementation. The efficient shape-sensing via the iFEM made it an attractive SHM technique, and soon, through introduction of various inverse elements, such as a four-node quadrilateral inverse shell element, known as iQS4 [33], its range of applications was expanded. In this context, successful implementations of the iFEM can be found in aerospace [34,35,36,37,38], marine [39,40], civil engineering [41], and machining [42] applications. More importantly, the theoretical basis of the iFEM has also experienced significant contributions through the works of various researchers [43,44]. One of the major theoretical contributions to the iFEM was made by Cerracchio et al. [45], who incorporated the refined zigzag theory [46] in the mathematical framework of the iFEM. Implementation of the RZT within the iFEM framework enabled accurate reconstruction of the displacement, section strains, and transverse shear strains through the thickness of the laminated materials, which was not possible according to the conventional theories such as first order shear deformation theory (FSDT). However, Kefal et al. [47] noticed that the governing mathematics shown in [45] can be further enhanced if new transverse shear strain terms would be added to the least-squares functional.

In addition to the mentioned advances achieved by the iFEM in terms of its efficiency and versatility, this method has also proven to be a robust damage-sensing and crack-monitoring tool. Colombo et al. [48] developed a damage anomaly based on the percent error difference between the in situ and reconstructed equivalent strain measure. In another study, Colombo et al. [49] enhanced the anomaly index in [48] and verified the iFEM-based damage detection analysis with experimental measurements. In the same context, Abdollahzadeh et al. [50] defined a damage index based on the difference between the equivalent von Mises strains obtained from the reconstructed damaged strain field and measured intact strain field, and assessed the performance of several inverse element types to identify damage in curved shell structures. Additionally, Li et al. [51] proposed a strain-based damage index associated with vibration modes, and performed damage detection analysis of a cantilevered plate with single and multiple damage cases. This study was complemented by integrating a convolutional neural network within the iFEM framework, which resulted in accurate damage identification and quantification [52]. Li et al. [53] demonstrated the robustness of iFEM/iQS4 in delamination detection by developing a damage index based on a pseudo-excitation method. The iFEM has also been exploited for crack detection and monitoring applications. In this regard, Kefal et al. [54] incorporated the peridynamic theory within the iFEM and proposed a robust real-time crack prognosis system. In another study, Oboe et al. [55] introduced a “Gaussian likelihood index” for estimating the size of the crack under mode I failure. To this end, several iFEM models associated with different damage scenarios were developed, and through calculating the likelihood index, the length of the crack can be approximated. The findings of this research effort were further consolidated by validating the novel methodology against the experimental test data. Most recently, Roy and Gherlone [56] targeted delamination damage detection in composite structure. In their study, the unidirectional strain data acquired from the composite material were smoothed over the entire problem domain, which were then used to locate delamination damage. However, they highlighted that accurate detection of damage requires collecting strain data from the vicinity of the delaminated region. Additionally, their study solely focused on identifying the in-plane position of the damage (without the fine detail of the through-the-thickness location of the delaminated region). More recently, both in-plane and through-the-thickness localization of the delamination was achieved based on iFEM–RZT formulation proposed by Ganjdoust et al. [57]. The superior capability of this recent damage detection methodology was demonstrated for static loading conditions only.

To the best of the authors’ knowledge, the iFEM has never been implemented to monitor delamination damage in composite plates subjected to dynamic vibrations. The present research effort aims to fill such an important research gap by devising a robust damage diagnosis system based on the iFEM–RZT methodology for the first time in the literature. The formulation proposed in the present study facilitates delamination detection in thin vibrating laminated composite structures and involves (i) identifying the onset of delamination damage, (ii) locating the in-plane and through-the-thickness position of the delamination, and (iii) reconstructing the shape of the delaminated region. This approach not only alleviates the need for an accurate and efficient damage monitoring system for composite materials but is also very versatile in terms of implementation. Furthermore, the novel features of the toolbox make it an attractive tool for studying and understanding the dynamic behavior and randomness of the delamination damage in real time.

The paper is structured in the following order: In Section 2, the governing mathematics of the iFEM–RZT within the context of the iRZT4 inverse element [58] is presented, and the iFEM–RZT-based damage detection toolbox is reviewed. Section 3 includes numerical case studies. In this section, the results of in-plane/through-the-thickness damage detection are presented, and a detailed interpretation associated with the results from each numerical case study is presented. Finally, based on the findings of the study, concluding remarks are stated in Section 4.

## 2. Theoretical Framework

The damage detection strategy was established within the framework of the iFEM–RZT. The mathematics of this delamination detection strategy were developed using the iRZT4 inverse-shell element. Then, damage indicators were calculated utilizing the reconstructed strain field. This section includes the underlying theories of the iFEM–RZT algorithm, and the equivalent strain-based damage detection method.

### 2.1. RZT and Derivatization of the iRZT4 Inverse Element

The structure is studied from the perspective of a (i) fixed global coordinate system, (X,Y,Z), (ii) a local coordinate system, $\left(x,z)\equiv (x_{1},x_{2},z\right)$, and (iii) an isoparametric coordinate system, denoted by (s,t). Consider that a laminated composite shell (Figure 1a) consists of N layers, the displacements of a given material point at the kth layer of the laminate can be obtained based on the RZT, using the following set of kinematic relations:(1a)$$u_{1}^{\left(k\right)}(x,z)=u_{1}^{\;}(x)+z\theta_{2}(x)+\phi_{1}^{\left(k\right)}(z)\psi_{2}(x)$$
(1b)$$u_{2}^{\left(k\right)}(x,z)=u_{2}(x)-z\theta_{1}(x)-\phi_{2}^{\left(k\right)}(z)\psi_{1}(x)$$
(1c)$$u_{z}(x)=w(x)$$
where $\left(u_{1},u_{2},u_{z}\right)$ show translations, $\left(\theta_{1},\theta_{2}\right)$ show bending rotations, and $\left(\psi_{1},\psi_{2}\right)$ show the zigzag rotations of a given material point within the plate. In fact, the zigzag rotations are deemed as the amplitude of the in-plane zigzag displacements $\phi_{1}^{\left(k\right)}(z)\psi_{2}(x)$ and $\phi_{2}^{\left(k\right)}(z)\psi_{1}(x)$, which are utilized to find an accurate distribution of the stresses and strains through the thickness of the laminated composite plate/shell. It must be noted that all the kinematic variables are functions of the local in-plane coordinates, $x=(x_{1},x_{2})$. Thus, the displacement vector for any material point in the laminated plate/shell will be given using the vector $u\equiv u(x)={\left\{\begin{array}{ccccccc} u_{1} & u_{2} & u_{z} & \theta_{1} & \theta_{2} & \psi_{1} & \psi_{2} \end{array}\right\}}^{T}$. Moreover, $\phi_{i}^{\left(k\right)}(z)$ (i=1,2), are piecewise linear functions to introduce the effects of zigzag rotations in the kinematic relations for any given material point (Figure 1b). The reason for defining these functions is that through-the-thickness displacement for each lamina/ply in a layered structure undergoes significant changes from one layer to another. On the other hand, zigzag functions secure the continuity of the displacements of each layer through the thickness of the laminate. For detailed information associated with the RZT and definition of the zigzag functions, interested readers are referred to Tessler et al. [46].

The analytical section strains and transverse shear strains for each ply are calculated using the RZT kinematic relations provided in Equation (1):(2a)$$\varepsilon^{\left(k\right)}\equiv \left\{\begin{array}{c} \varepsilon_{11}^{\left(k\right)}\\ \varepsilon_{22}^{\left(k\right)}\\ \varepsilon_{12}^{\left(k\right)} \end{array}\right\}=e(u)+z\kappa (u)+\mu^{\left(k\right)}(u,z)$$
(2b)$$\gamma^{\left(k\right)}\equiv \left\{\begin{array}{c} \gamma_{1z}^{\left(k\right)}\\ \gamma_{2z}^{\left(k\right)} \end{array}\right\}=H_{\gamma }^{\left(k\right)}(z)\gamma (u)+H_{\eta }^{\left(k\right)}(z)\eta (u)$$
where e(u) and κ(u) represent the membrane strain measure and bending curvatures, respectively. Moreover, $\mu^{\left(k\right)}(u,z)$ show the layer-wise zigzag strains. The explicit derivation of each of the mentioned strain measures is written as
(3a)$$e(u)={\left\{\begin{array}{ccc} u_{1,1} & u_{2,2} & u_{1,2}+u_{2,1} \end{array}\right\}}^{T}$$
(3b)$$\kappa (u)={\left\{\begin{array}{ccc} \theta_{2,1} & -\theta_{1,2} & -\theta_{2,2}+\theta_{1,1} \end{array}\right\}}^{T}$$
(3c)$$\mu^{\left(k\right)}(u,z)=H_{\phi }^{\left(k\right)}(z){\left\{\begin{array}{cccc} \psi_{2,1} & -\psi_{1,2} & \psi_{2,2} & -\psi_{1,1} \end{array}\right\}}^{T}$$

Here, $H_{\phi }^{\left(k\right)}(z)$ is defined through the thickness of the laminate for each layer as a function of the piecewise zigzag functions, $\phi_{i}^{\left(k\right)}$:(4)$$H_{\phi }^{\left(k\right)}=\left[\begin{array}{cccc} \phi_{1}^{\left(k\right)} & 0 & 0 & 0\\ 0 & \phi_{2}^{\left(k\right)} & 0 & 0\\ 0 & 0 & \phi_{1}^{\left(k\right)} & \phi_{2}^{\left(k\right)} \end{array}\right]$$

In addition, the components of the transverse shear strains can also be explicitly derived as
(5a)$$\gamma (u)\equiv {\left\{\begin{array}{cc} \gamma_{1} & \gamma_{2} \end{array}\right\}}^{T}={\left\{\begin{array}{cc} w_{,1}+\theta_{2} & w_{,2}-\theta_{1} \end{array}\right\}}^{T}$$
(5b)$$\eta (u)={\left\{\begin{array}{cc} \eta_{1}-\psi_{2} & \eta_{2}+\psi_{1} \end{array}\right\}}^{T}$$
where $H_{\gamma }^{\left(k\right)}$ and $H_{\eta }^{\left(k\right)}$ are given as
(6a)$$H_{\gamma }^{\left(k\right)}=\left[\begin{array}{cc} 1+\phi_{1,z}^{\left(k\right)} & 0\\ 0 & 1+\phi_{2,z}^{\left(k\right)} \end{array}\right]$$
(6b)$$H_{\eta }^{\left(k\right)}=\left[\begin{array}{cc} -\phi_{1,z}^{\left(k\right)} & 0\\ 0 & -\phi_{2,z}^{\left(k\right)} \end{array}\right]$$

The iRZT4 inverse-shell element (Figure 2) is developed by incorporating the RZT kinematics within the framework of the four-node quadrilateral inverse-shell element, iQS4. Using the isoparametric shape functions $N_{i}$, the position of a given point inside the 4-node inverse element can be approximated as
(7)$$x_{j}=\sum_{i=1}^{4}N_{i}x_{ji}\;(j=1,2)$$
with $x_{1i}$ and $x_{2i}$ showing the nodal coordinates of the inverse-shell element along their respective local coordinate axes.

Similarly, the components of the analytical displacements in Equation (1), can be obtained in an approximate form as
(8a)$$u_{1}=\sum_{i=1}^{4}N_{i}u_{1i}+\sum_{i=1}^{4}L_{i}\theta_{zi}$$
(8b)$$u_{2}=\sum_{i=1}^{4}N_{i}u_{2i}+\sum_{i=1}^{4}M_{i}\theta_{zi}$$
(8c)$$w=\sum_{i=1}^{4}N_{i}w_{i}-\sum_{i=1}^{4}L_{i}\left(\theta_{1i}-\psi_{1i}\right)-\sum_{i=1}^{4}M_{i}\left(\theta_{2i}-\psi_{2i}\right)$$
(8d)$$\theta_{j}=\sum_{i=1}^{4}N_{i}\theta_{ji}\;(j=1,2)$$
(8e)$$\psi_{j}=\sum_{i=1}^{4}N_{i}\psi_{ji}\;(j=1,2)$$
where $L_{i}$ and $M_{i}$ are higher order anisoparametric shape functions, which are defined in terms of the bilinear shape functions $N_{i}$. Additionally, $\theta_{zi}$ and $\psi_{zi}$ are the artificial drilling rotation and artificial zigzag rotation degrees of freedom, which guarantee avoiding singular solutions. As a result of discretizing the problem domain, the numerical counterparts of analytical strain measures can be recovered in terms of the element displacement vector $u^{e}$ according to the kinematic relations of the iRZT4 element as
(9)$$\chi (u^{e})=B^{\chi }u^{e}\;(\chi =e,\kappa ,\mu ,\gamma ,\eta )$$
where the nodal degrees of freedom for the ith node of an individual iRZT4 element are given as
(10)$$u_{i}^{e}={\left\{\begin{array}{ccccccccc} u_{1i} & u_{2i} & w_{i} & \theta_{1i} & \theta_{2i} & \theta_{zi} & \psi_{1i} & \psi_{2i} & \psi_{zi} \end{array}\right\}}^{T}\;(i=1,2,3,4)$$

Within the framework of the iFEM–RZT, the least-squares functional accounts for five distinct deformation fields. In this context, for each field, a function is defined in terms of the difference between the numerical strain measures (Equations (3) and (5)) and the in situ strain measurements. This can be defined as
(11a)$$\phi_{e}(u^{e})=e(u^{e})-E$$
(11b)$$\phi_{\kappa }(u^{e})=\kappa (u^{e})-K$$
(11c)$$\phi_{\mu }(u^{e},z_{j})=\mu^{\left(k\right)}(u^{e},z_{j})-M^{j}$$
(11d)$$\phi_{\gamma }(u^{e})=\gamma (u^{e})-\Gamma$$
(11e)$$\phi_{\eta }(u^{e})=\eta (u^{e})-H$$

Equations (11a–c) refer to the membrane strains, bending curvatures, and zigzag strains, respectively, while Equations (11d–e) correspond to the transverse shear strain measures. Additionally, the experimental strain measures are given as E for membrane strain measures, K for bending curvatures, $M^{j}$ for zigzag section strains at the jth interface of the laminate, Γ and H for the transverse shear strains.

As is illustrated in Figure 3, the experimental strain data can be acquired from the bounding surfaces, and through the thickness of the shell. In the Figure, $\varepsilon_{i}^{+}$, $\varepsilon_{i}^{-}$, and $\varepsilon_{i}^{j}$ refer to section strain measurements of the ith sensor, at the top surface, bottom surface, and jth interface of the laminate, respectively. The section strain measures E, K, and $M^{j}$ can be obtained by manipulating the in situ strain measures, as is outlined in [58]. However, measuring the experimental strains associated with the transverse shear strain terms Γ and H is not as straightforward as the strain measures described previously, and they require auxiliary techniques such as smoothed iFEM [44] to be obtained. Nevertheless, the effects of the transverse shear strain terms are usually negligible in thin shells in comparison to other experimental strain data.

The least-squares functional can be written for a single iRZT4 element by integrating the weighted summation of the norm of the difference functions (Equation (11)) over the area of the inverse element as
(12)$$\Phi (u^{e})=\frac{1}{A_{e}}\iint_{A_{e}}\left[\begin{array}{l} w_{e}{\left\|\phi_{e}(u^{e})\right\|}^{2}+w_{\kappa }{\left\|\phi_{\kappa }(u^{e})\right\|}^{2}+w_{\mu }{\left\|\phi_{\mu }(u^{e},z_{j})\right\|}^{2}\\ \;\;\;\;\;\;\;\;\;\;\;\;\;\;\;\;\;\;\;\;\;\;\;\;\;\;\;\;\;+w_{\gamma }{\left\|\phi_{\gamma }(u^{e})\right\|}^{2}+w_{\eta }{\left\|\phi_{\eta }(u^{e})\right\|}^{2} \end{array}\right]dx_{1}dx_{2}$$

In this equation, ${\left\|X\right\|}^{2}$ is the normal second order Euclidean norm operator, which can be determined by calculating the dot product X⋅X. Moreover, $w_{\chi }$ with χ={e,κ,μ,γ,η} are the scalar weight coefficients that are utilized to enforce the effects of strain data for a certain element. In this regard, $w_{e}$, $w_{\kappa }$, and $w_{\mu }$ are associated with the membrane, bending, and zigzag strain measures, respectively, while $w_{\gamma }$ and $w_{\eta }$ are related to the transverse shear strains. These coefficients take a value of $w_{\chi }=1$ if the sensor data are available at a certain location, and if no experimental data are available, a very small number $w_{\chi }=\alpha$ relative to unity is assigned to them, e.g., $\alpha =10^{-5}$. As a result, the elements for whom the sensor data are available will have a more dominant role in the reconstruction of the displacement field. By minimizing the least-squares functional subject to the local displacement vector, a system of equations can be obtained at the element level:(13)$$\frac{\partial \Phi (u^{e})}{\partial u^{e}}=0\;\;\;\to \;\;\;k^{e}u^{e}-f^{e}=0$$
in which $k^{e}$ denotes the local (element) iFEM matrix, and $f^{e}$ is the local (element) right-hand-side (RHS) vector.
(14a)$$k^{e}=\frac{1}{A_{e}}\iint_{A_{e}}\left[\begin{array}{l} w_{e}{\left(B^{e}\right)}^{T}B^{e}+{\left(2h\right)}^{2}w_{\kappa }{\left(B^{\kappa }\right)}^{T}B^{\kappa }+\\ w_{\mu }{\left(H_{\phi }^{\left(k\right)}(z_{j})B^{\mu }\right)}^{T}H_{\phi }^{\left(k\right)}(z_{j})B^{\mu }+\\ w_{\gamma }{\left(B^{\gamma }\right)}^{T}B^{\gamma }+w_{\eta }{\left(B^{\eta }\right)}^{T}B^{\eta } \end{array}\right]dx_{1}dx_{2}$$



(14b)
$$f^{e}=\frac{1}{A_{e}}\iint_{A_{e}}\left[\begin{array}{l} w_{e}{\left(B^{e}\right)}^{T}E+{\left(2h\right)}^{2}w_{\kappa }{\left(B^{\kappa }\right)}^{T}K+\\ w_{\mu }{\left(H_{\phi }^{\left(k\right)}(z_{j})B^{\mu }\right)}^{T}M^{j}+\\ w_{\gamma }{\left(B^{\gamma }\right)}^{T}\Gamma +w_{\eta }{\left(B^{\eta }\right)}^{T}H \end{array}\right]dx_{1}dx_{2}$$



Like the (forward) FEM, the local iFEM matrix and the local RHS vector are transformed from the local coordinates to the global coordinates. Afterwards, they are assembled into a global system of equations. In this regard, it can be written that:(15a)$$K_{G}=\sum_{i=1}^{n_{el}}{\left(T^{e}\right)}^{T}k^{e}T^{e}$$
(15b)$$F_{G}=\sum_{i=1}^{n_{el}}{\left(T^{e}\right)}^{T}f^{e}$$
where $n_{el}$ is used to denote the total number of the inverse elements. In addition, $K_{G}$ and $F_{G}$ are the global iFEM matrix and global RHS vectors, respectively, and $T^{e}$ is the transformation matrix from the local reference frame to the global coordinates. In addition, the global displacement vector is also given as
(16)$$U_{G}=\sum_{i=1}^{n_{el}}{\left(T^{e}\right)}^{T}u^{e}$$

Hence, the global system of equations will be defined as
(17)$$K_{G}U_{G}=F_{G}$$

To solve the given global system of equations, the rows and columns associated with the essential boundary conditions are reduced, and a new reduced global system of equations is obtained. Then, the reduced displacement vector is obtained using matrix algebra.
(18)$$K_{R}U_{R}=F_{R}\;\;\;\to \;\;\;U_{R}={\left(K_{R}\right)}^{-1}F_{R}$$

Thus, the displacement field, and consequently the strain field of the laminated shell structures, can be recovered utilizing the iFEM–RZT methodology.

### 2.2. Damage Detection Toolbox Based on iFEM–RZT

The equivalent strain-based damage detection toolbox is developed using the shape-sensing results of the iFEM analysis. Based on the reconstructed strain field, various damage indices are defined. These damage indices enable the iFEM–RZT methodology to detect the in-plane location and configuration of the delamination damage. Moreover, they identify the location of the delamination damage through the thickness of the layered shell. The robustness and accuracy of the shape-sensing results via the iFEM ensures the accuracy of the damage detection via the present methodology. Furthermore, it is known that in shape-sensing via the iFEM, no a priori information regarding the material model or loading conditions is required; however, the use of RZT in this delamination damage detection toolbox requires information about the material model so that the through-the-thickness location of delamination is identified.

The damage detection toolbox based on iFEM–RZT uses equivalent von Mises strains to calculate damage indices. To find the in-plane location of the damage, the equivalent von Mises strain associated with membrane strain measure and bending curvature are calculated. To find the equivalent strain measures, the first and second principal strains for membrane and bending strain measures are established as
(19)$$\varepsilon_{1,2}^{\chi }=\frac{\chi_{11}+\chi_{22}}{2}\pm \sqrt{{\left(\frac{\chi_{11}-\chi_{22}}{2}\right)}^{2}+{\left(\frac{\chi_{12}}{2}\right)}^{2}}\;(\chi =e,\kappa )$$
where $e={\left\{\begin{array}{ccc} e_{11} & e_{22} & e_{12} \end{array}\right\}}^{T}$ and $\kappa ={\left\{\begin{array}{ccc} \kappa_{11} & \kappa_{22} & \kappa_{12} \end{array}\right\}}^{T}$ are used to denote the reconstructed membrane strain measures and bending curvatures, respectively, and the equivalent von Mises strain associated with each strain measure will be obtained as
(20)$$\varepsilon_{VM}^{\chi }=\sqrt{{\left(\varepsilon_{1}^{\chi }\right)}^{2}-\varepsilon_{1}^{\chi }\varepsilon_{2}^{\chi }+{\left(\varepsilon_{2}^{\chi }\right)}^{2}}\;(\chi =e,\kappa )$$

As a result, the membrane damage index, $D_{e}$, and the twisting damage index, $D_{\kappa }$, can be formulated as
(21)$$D_{\chi }(\varepsilon_{VM}^{\chi })=\left|\frac{{\left(\varepsilon_{VM}^{\chi ,B}\right)}_{i}-{\left(\varepsilon_{VM}^{\chi }\right)}_{i}}{\varepsilon_{VM}^{\chi ,\max}}\right|\;(\chi =e,\kappa ;i=1,2,\ldots ,n_{el})$$

Here, $\varepsilon_{VM}^{\chi ,\max}$ is the maximum reconstructed von Mises strain using the iFEM, and $\varepsilon_{VM}^{\chi ,B}$ shows the baseline value for the equivalent strain measures at each mode. This reference value is associated with the last instant when the structure is intact. The baseline value is updated consistently until the first signs of von Mises strain localization are observed. In this case, the time increment associated with the onset of damage localization will be the final baseline measurement until the end of the damage detection analysis.

Using a similar process, the location of the delaminated region through the thickness of the shell structure can be determined. In this context, the principal strains for each layer are first calculated as
(22)$$\varepsilon_{1,2}^{\left(k\right)}=\frac{\varepsilon_{11}^{\left(k\right)}+\varepsilon_{22}^{\left(k\right)}}{2}+\sqrt{{\left(\frac{\varepsilon_{11}^{\left(k\right)}-\varepsilon_{22}^{\left(k\right)}}{2}\right)}^{2}+{\left(\frac{\gamma_{12}^{\left(k\right)}}{2}\right)}^{2}}$$
using which, the average equivalent strains for each layer are obtained as
(23)$${\bar{\varepsilon }}_{VM}^{\left(k\right)}=\frac{1}{2h^{\left(k\right)}}\int_{z_{k-1}}^{z_{k}}\sqrt{{\left(\varepsilon_{1}^{\left(k\right)}\right)}^{2}-\varepsilon_{1}^{\left(k\right)}\varepsilon_{2}^{\left(k\right)}+{\left(\varepsilon_{1}^{\left(k\right)}\right)}^{2}}dz$$
and the damage indices for each lamina, $D_{\left(k\right)}$, will be given as
(24)$$D_{\left(k\right)}({\bar{\varepsilon }}_{VM}^{\left(k\right)})=\left|\frac{{\left({\bar{\varepsilon }}_{VM}^{(k),B}\right)}_{i}-{\left({\bar{\varepsilon }}_{VM}^{\left(k\right)}\right)}_{i}}{{\bar{\varepsilon }}_{VM}^{(k),\max}}\right|\;(i=1,2,\ldots ,n_{el})$$
where, ${\bar{\varepsilon }}_{VM}^{(k),\max}$ is the maximum reconstructed average von Mises strain of each ply, and ${\bar{\varepsilon }}_{VM}^{(k),\max}$ is the reference average strain measure for each lamina. Hence, using the membrane and twisting damage indices, the in-plane location and shape of the delaminated region is configured, and then, a layer-wise inspection of delamination damage is performed, to identify the through-the-thickness position of the delaminated region in the shell. For a detailed account of the implementation of the present delamination damage detection method, see [57].

## 3. Numerical Examples

The robustness and efficiency of the proposed damage detection toolbox has been demonstrated in several numerical case studies in the previous study by the same authors, all of which included time-invariant loading. However, complex engineering structures available in aerospace and marine industries often undergo dynamic loads, and the loading condition might be of a harmonic, stochastic, or random nature. In addition, it is known that structural damage in composite materials is a progressive process. Therefore, damage detection systems for composite materials must be capable of identifying the onset of damage, and providing information regarding the behavior of the damage, such as its intensity over time and load steps.

In the present effort, two numerical case studies were investigated, namely a cantilevered plate subject to harmonic load and a cantilevered curved composite shell subject to random bending load. In both examples, the efficacy of the inverse damage detection strategy is studied in terms of identifying the initiation of the delamination damage, and the ability of the method in locating the in-plane and through-the-thickness position and configuration of the delamination damage. It is also worth mentioning that, in neither of the test cases, the damage model is natural, i.e., damage is not modeled in a progressive manner. Indeed, it is assumed that the shell structure is in a pristine state until a certain time increment, after which delamination damage appears in the structure. Therefore, it is worth noting that the present numerical examples do not entail this damage detection technique to monitor the propagation and growth of the delaminated region. Herein, only the initiation and intensity of the delamination damage are monitored over time.

### 3.1. Cantilevered Plate Subject to Harmonic Load

In this example, a cantilevered laminated composite plate is considered. The loading condition of the problem is defined according to the vibrational characteristics of the plate. In other words, if the kth mode of vibration is bending, the harmonic load will also be a bending load. Given this, the first two natural frequencies and mode shapes of the intact system were determined using the commercial FE software, Ansys Mechanical APDL. Based on the results, it can be seen that the first vibration mode of the system is bending (first flap) with the fundamental natural frequency determined as $\omega_{n_{1}}=19.3422$ rad/s, whereas the second mode corresponds to torsion (first torsion) and its respective natural frequency is $\omega_{n_{2}}=37.9360$ rad/s. Hence, the present example consists of two sub-problems:Cantilevered laminated composite plate subject to harmonic bending load;Cantilevered laminated composite plate subject to harmonic torsion load.

Figure 4 shows the geometry of the plate, where the in-plane and through-the-thickness configurations of the delaminate region are also depicted. The plate has a length and width of 1000 mm and a thickness of 4 mm. Hence, it can be categorized as a thin plate (Span-to-thickness ratio 1000/4=250). The stacking sequence of the laminated plate is given by [0/±45/90] and the orthotropic material properties for each lamina are provided in Table 1.

On the other hand, Figure 5 shows the loading conditions for each of the sub-problems mentioned above. The figure shows that, in each case, the harmonic load is applied to the free edge of the plate.

In addition, the delaminated region is a 50×50 square region, which is located at the interface of the 3rd and 4th layers. At the intact cross-section, all the laminae have the same thickness, while at the defected cross-section, the thickness of the 3rd and 4th plies is reduced to 95% of their intact thickness; therefore, the thickness of the delamination damage is 10% of the total thickness of the laminate. In fact, the delaminated region is simply modeled as a lack of layer in the forward analysis. In both sub-problems of this case study, harmonic load is applied over a period of 4 s, and after t=1 s, delamination damage is activated in the plate.

Moreover, the input strain data for the iFEM–RZT algorithm are collected from the top/bottom surfaces of the laminated plate in the present case study. In addition, through-the-thickness strain data from the interface of the 1st and 2nd layers are also acquired. As it was discussed in Section 2.2, the in situ strain data include all the components of the section strains, ${\left\{\begin{array}{ccc} \varepsilon_{11} & \varepsilon_{22} & \gamma_{12} \end{array}\right\}}^{T}$. These data are collected according to the sensor placement scheme depicted in Figure 6.

#### 3.1.1. Cantilevered Laminated Composite Plate Subject to Harmonic Bending Load

As it is illustrated in Figure 5a, the harmonic distributed bending load is applied to the free edge of the composite plate. The bending load is given as $F_{b}(t)=F_{b0}\sin(\omega_{1}t)$, in which the amplitude of the force is $F_{b0}=100$ N and the excitation frequency is defined as $\omega_{1}=0.95\omega_{n_{1}}$. The value of the excitation frequency was selected close to the natural frequency to trigger the resonance in the first vibrational mode of the plate. Strain data were collected from the plate with a frequency of 100 Hz, according to the sensor placement scheme depicted in Figure 6.

The strain field of the plate was reconstructed through the iFEM–RZT algorithm. In this context, the weight coefficients used in the iFEM–RZT least-squares functional (Equation (12)) were set equal to unity for the iRZT4 elements with sensor data, whereas, for elements that do not possess sensor data, their value was $w_{\chi }=10^{-4}$ for χ={e,κ,μ,γ,η}. Delamination detection was performed by first determining the in-plane location of the damage. Hence, first, the membrane and twisting damage indicators are calculated. The contour plots associated with the membrane and twisting damage indices are presented in Figure 7 and Figure 8, respectively. It is evident that, prior to damage initiation, neither of the damage indices showed any sign of damage localization in the plate. However, after the damage was initiated, both the membrane and twisting damage indices identified a damaged region at the location of the damage, according to Figure 4. The magnitude of the damage at this stage was not significantly high; however, as the harmonic loading continues over time, at t=3.5 s, both damage indicators show the delamination damage at its highest intensity. Moreover, using the contour plots provided in Figure 7 and Figure 8, a rough estimation of the in-plane shape of the damage is also enabled.

Additionally, the location of the delaminated region can be determined through a layer-wise inspection of damage based on the proposed strategy. In this context, after finding the in-plane location of the damage, at each time increment, damage indices for each layer are calculated via Equation (24) at the delaminated region. The variations of the normalized damage identifiers are plotted and illustrated in Figure 9.

According to these results, it can be seen that the damage indices associated with the 3rd and 4th layers acquire values greater than those related to the other layers. These results imply that the delaminated region is located between the 3rd and 4th layers. Furthermore, the values of the damage identifiers change drastically after t=1 s, indicating the onset of delamination damage. Additionally, from the variations in the layer-wise damage indices in Figure 9, the effects of resonance on the intensity of the damage can be seen.

#### 3.1.2. Cantilevered Laminated Composite Plate Subject to Harmonic Torsion Load

As was mentioned earlier, the plate undergoes torsional response in its second mode of vibration. In this sub-problem, the present damage detection strategy is implemented to identify delamination damage when the structure is subject to harmonic torsion load. In this regard, the free edge of the plate is subject to a harmonic torsional load (Figure 5b) given by $F_{t}(t)=F_{t0}\sin(\omega_{2}t)$. The amplitude of the force is $F_{t0}=100$ N, and the excitation frequency is related to the second natural frequency of the plate through the relation $\omega_{2}=0.95\omega_{n_{2}}$. In contrast to the previous example, strain data acquisition is performed with a frequency of 200 Hz throughout the entire analysis time.

Like the previous example, the membrane and twisting damage indices are calculated using Equation (21), and the results are demonstrated in Figure 10 and Figure 11. According to these results, the iFEM–RZT-based delamination detection approach is able to detect the in-plane location of the damage in the structure and distinguish the intact and damaged states of the laminated plate. In addition, it can be observed that, at various time increments, the intensity of the damage changes. In contrast to the previous case, the values of the membrane and twisting damage indices never reach the maximum intensity of the damage, i.e., unity. The correlation between this phenomenon and possible effects of increasing the frequency of the vibration requires further investigation; nevertheless, the proposed damage detection method is still capable of finding the in-plane location of the damage and approximating its morphology.

The results of the through-the-thickness damage detection are presented in Figure 12. Like the previous case, the layer-wise damage identifiers correctly show the interface of the 3rd and 4th layers as the through-the-thickness location of the delaminated region, as the normalized layer-wise damage indices show more intense levels of damage for these two layers. Additionally, the resonant behavior of the damage is projected in the presented plot, and the variations in the values of the layer-wise damage indices shows traces of beating phenomenon, which is commonly encountered in resonant systems.

In general, the results of the two sub-problems show that delamination damage can be identified effectively, in terms of its exact in-plane and through-the-thickness location in a thin vibrating laminated composite shell. Moreover, the shape of the delaminated region is reconstructed using the membrane and twisting damage indices, and the resonant effects of the damage are highlighted in the variations of the layer-wise damage indices with respect to time.

### 3.2. Cantilevered Curved Shell Subject to Random Bending Load

Complex structures, commonly used in aerospace or marine industries, are often subject to harsh and random loading conditions. In this subsection, the damage detection method is implemented to locate delamination damage in a composite curved shell subjected to random distributed bending load (Figure 13). The length of the shell is 200 mm, and the curvature of the shell has a radius of 100 mm. Additionally, the total thickness of the shell is 3.5 mm, and it comprises seven equally thick layers of unidirectional carbon–epoxy composite (Table 1). The stacking sequence of the laminate is given as [0,90,0,90,0,90,0].

The structure is subject to a random distributed bending load from its free edge that varies between 0 N and 200 N over a period of 30 s. In addition, delamination damage is located at the interface of the 1st and 2nd plies, which is initiated at t=6 s. It is assumed that, because of the delamination, the thickness of the 1st and 2nd laminae decrease by 10% of their undamaged thickness, indicating that the delamination has a thickness equal to 0.1 mm. The in situ strain data are acquired from the top/bottom surfaces of the shell, as well as from the interface of the 6th and 7th plies, according to the sensor placement scheme illustrated in Figure 14. The figure also shows the in-plane location of the rectangular delaminated region.

Figure 15 shows the variations in the magnitude of the random bending load with respect to the time of the analysis. The magnitude of the bending load for each time step has been provided in Appendix A. The iFEM–RZT algorithm processes the strain data associated with each load step to reconstruct the strain field of the curved shell. In this regard, the weight coefficients, $w_{\chi }$, are defined exactly as in the previous case study, i.e., $w_{\chi }=1$ for elements with sensors and $w_{\chi }=10^{-4}$ for elements without sensors, so that elements with sensors will have a more dominant effect on reconstructing the displacement and strain fields.

Considering that the structure undergoes greater levels of strain as the magnitude of the random load is higher, the in-plane damage detection results have been provided for the load steps with higher magnitudes of load and the time step corresponding to the instant before the initiation of the damage. Figure 16 and Figure 17 show the membrane and twisting damage indices associated with these time steps for the curved structure.

The contour plots for the membrane and twisting damage indices show accurate identification of the in-plane location and shape of the delaminated region when the structure is subject to extreme changes in the applied load. Intuitively, as the magnitude of the load increases, it is expected that the structure experiences intense levels of damage, or extreme changes of the equivalent von Mises strains from two consecutive load steps might make the damage detection prone to error; nevertheless, the results show that, prior to damage initiation, higher values for the random bending load do not have any effect on the performance of the proposed damage detection methodology. Indeed, the iFEM-based damage monitoring system provides an accurate distinction between the pristine and damaged state of the structure. In addition, slight variations in the maximum load magnitudes are taken into account by the present damage detection approach. These effects are reflected in the varying intensity of the damage indices derived for various loads, indicating that the iFEM–RZT-based model is sensitive to the load variations. This robustness can be credited to accurate reconstruction of the displacement and strain fields achieved via the iFEM. In addition, it must be noted that, although the structure does not experience damage until t=5 s, the values for damage indices are not equal to zero. Nonetheless, these results can be interpreted through continuum damage models. As an example, the acceptable range for the critical value of the damage index for metallic structures is usually $0.2<D_{c}<0.8$, with $D_{c}$ denoting the critical damage index [59]. The correlation between the presented results and the critical damage range for composite materials can be the topic of future research efforts.

Furthermore, to locate the through-the-thickness position of the delaminated region, layer-wise damage indices were calculated using Equation (24). After identifying the in-plane location of the damage utilizing the membrane and twisting damage indices, the layer-wise damage identifiers can be calculated at the delaminated region. The damage indices calculated at the damaged region for each ply are depicted in Figure 18.

According to these curves, the layer-wise damage indices identify the through-the-thickness position of the delaminated region as the interface of the 1st and 2nd plies. However, the peaks of the curves illustrated in Figure 18 are not necessarily compatible with the peak values of the membrane and twisting damage indices. Nevertheless, they can still be used for detecting the position of the defected region through the thickness of the curved structure.

In summary, it was demonstrated that the proposed damage detection strategy identified the configuration of the damage effectively in the curved laminated shell. If the time increments associated with the most intense levels of damage are to be considered critical time steps, the results of the in-plane damage detection are more reliable than the through-the-thickness damage inspection. Nonetheless, in terms of providing the location of the delaminated region within the structure, the robustness of the delamination detection method is guaranteed. In general, the delamination detection and damage monitoring results of the proposed methodology show superiority to the results available in the literature. Several features make the proposed iFEM–RZT-based technique a valuable tool. Among these features, the capability of the present approach to accurately locate the delaminated region is one of the most important features, which is a rather far-fetched objective when using VSHM techniques [25]. On the other hand, some of the delamination detection techniques, such as the ones proposed in [23,24], have tried to alleviate this issue of the VSHM methods in beam problems; however, to the best of the authors’ knowledge, their methodology has not been implemented to problems involving plates/shells. Moreover, vibrational modes will become more complex as the dimensions of the domain increase, whereas the iFEM–RZT-based method is very versatile in terms of numerical/practical implementation thanks to its well-developed library of inverse elements [33,44,47,50]. Finally, the most recent iFEM-based delamination detection technique [56] is more economical than the present methodology in terms of instrumentation; however, it lacks the capability to detect the delaminated region through the thickness of the laminate, whereas the technique proposed in the present paper is exempt from such a restriction.

## 4. Concluding Remarks

In this paper, a robust damage monitoring technique is presented for delamination detection in composite structures subjected to forced vibrations. To illustrate the efficiency of the proposed methodology, two numerical case studies have been conducted. In the first example, the harmonic vibrations of a cantilevered composite plate were considered, where the applied harmonic load was closely related to the vibrational modes of the composite plate. In the second example, the random vibration of a cantilevered curved composite shell was studied. According to the results of both examples, the iFEM–RZT-based delamination detection strategy was found to be an effective tool in terms of diagnosing damage in vibrating composite shells. In this context, it has been demonstrated that, by utilizing this technique, valuable information associated with (i) the existence and initiation of delamination damage in the laminate, (ii) the in-plane and through-the-thickness position of the damage, and (iii) the shape and extent of the delaminated region can be obtained by using only sensor data (i.e., without any material/loading information of the laminate). Moreover, it is shown that this methodology is immune to load variations in the case of random vibrations. Furthermore, the effects of resonance of the system in harmonic vibrations can be captured using this technique. Overall, it has been proven that the iFEM–RZT damage detection method is a viable technique for the real-time diagnosis of delamination within thin complex/curved/built-up structural topologies made of laminated composites.

As a limitation of the proposed approach, it is only applicable to laminated structures (i.e., not directly for 3D woven composites). Additionally, during the analysis, the present damage detection approach has only been applied to problems that involve thin plates/shells. Therefore, these topics can be the subject of future research work to further enhance the present damage detection strategy’s performance. Additionally, the future direction of this research could involve implementing the present methodology to monitor the time-dependent variation and propagation of the delaminated damage and the study of multiple delaminated regions in the structure.

## Figures and Tables

**Figure 1 sensors-23-07926-f001:**
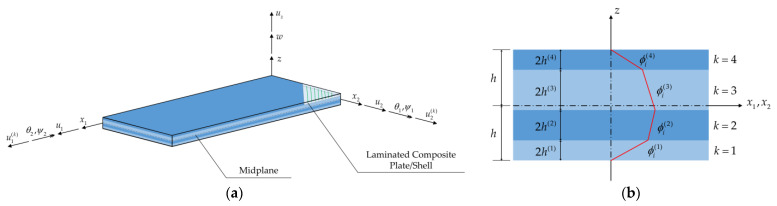
(**a**) Laminated composite shell; (**b**) Zigzag functions defined through the thickness of the laminated shell.

**Figure 2 sensors-23-07926-f002:**
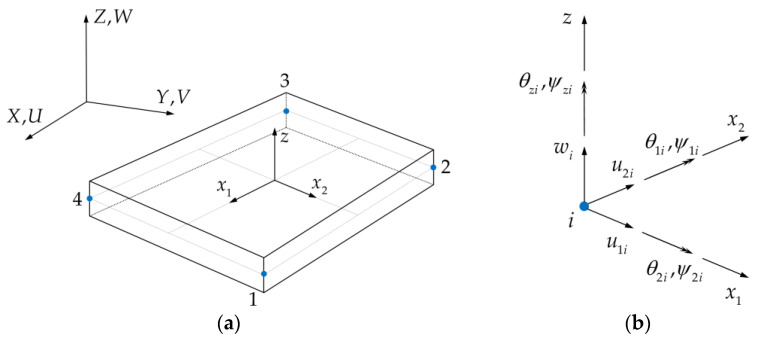
Inverse shell element, iRZT4: (**a**) from the perspective of the global and local coordinate systems; (**b**) nodal degrees of freedom for node *i* of the inverse element.

**Figure 3 sensors-23-07926-f003:**
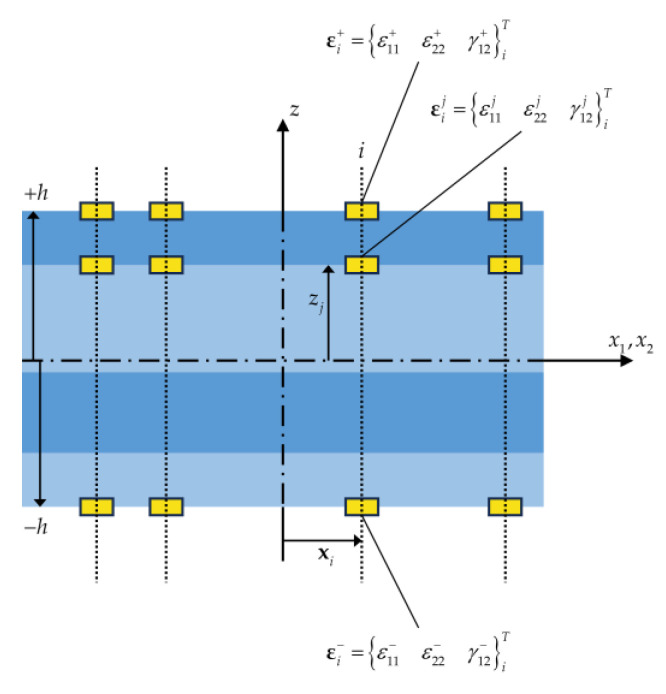
In situ strain measurements from the top/bottom surfaces and through the thickness of the laminated shell.

**Figure 4 sensors-23-07926-f004:**
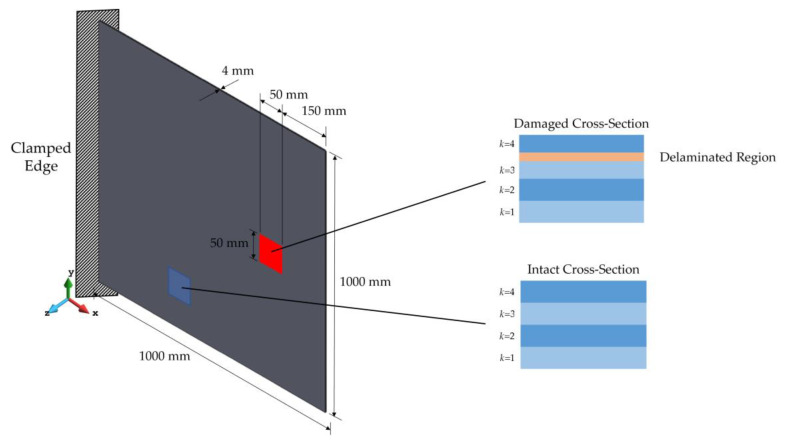
The geometry of the delaminated cantilevered laminated composite plate.

**Figure 5 sensors-23-07926-f005:**
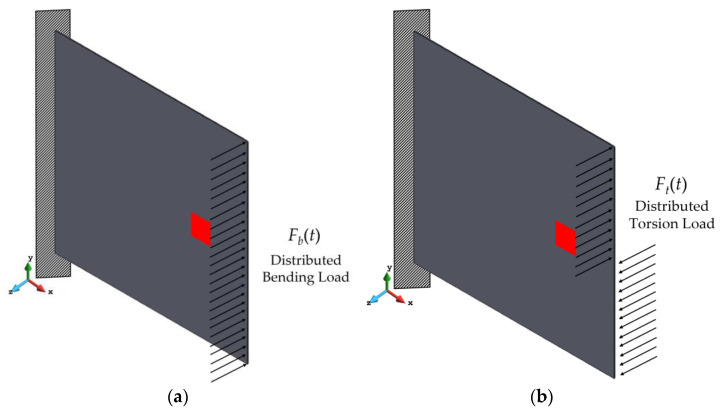
Cantilever laminated composite plate subject to (**a**) harmonic bending load, (**b**) harmonic torsion load.

**Figure 6 sensors-23-07926-f006:**
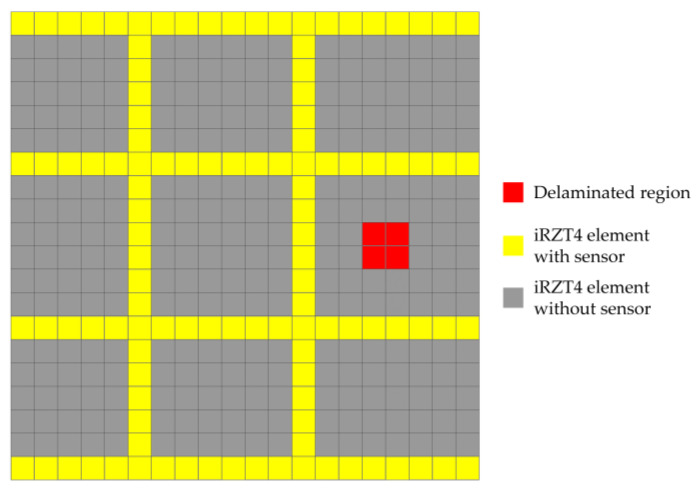
Sensor placement pattern for the cantilevered laminated composite plate.

**Figure 7 sensors-23-07926-f007:**
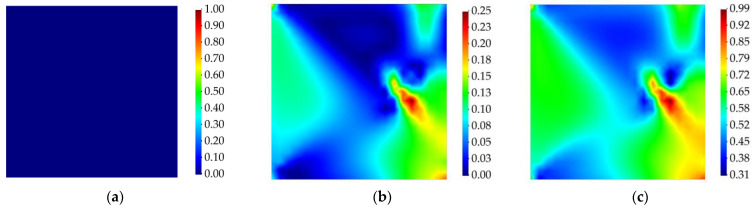
Membrane damage index for cantilevered laminated composite plate subject to harmonic bending load; (**a**) the instant prior to damage initiation, t=1 s, (**b**) the instant of damage initiation, t=1.01 s, (**c**) the instant with the maximum damage, t=3.5 s.

**Figure 8 sensors-23-07926-f008:**
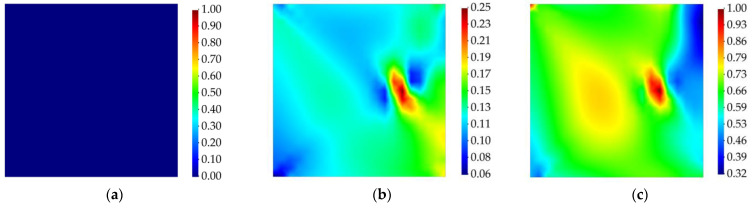
Twisting damage index for cantilevered laminated composite plate subject to harmonic bending load; (**a**) the instant prior to damage initiation, t=1 s, (**b**) the instant of damage initiation, t=1.01 s, (**c**) the instant with the maximum damage, t=3.5 s.

**Figure 9 sensors-23-07926-f009:**
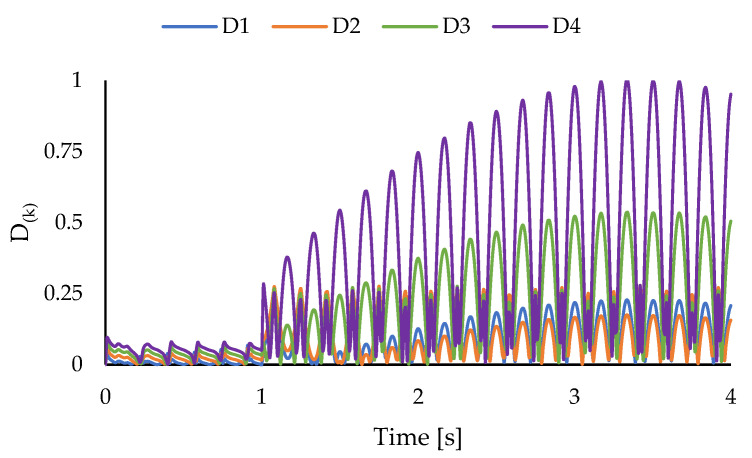
Layer-wise damage indices for cantilevered plate under harmonic bending load.

**Figure 10 sensors-23-07926-f010:**

Membrane damage index for cantilevered laminated composite plate subject to harmonic torsion load; (**a**) the instant prior to damage initiation, t=1 s, (**b**) the instant of damage initiation, t=1.005 s, (**c**) the instant with the maximum damage, t=1.7 s.

**Figure 11 sensors-23-07926-f011:**
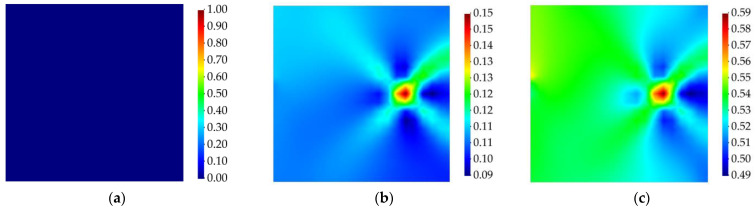
Twisting damage index for cantilevered laminated composite plate subject to harmonic torsion load; (**a**) the instant prior to damage initiation, t=1 s, (**b**) the instant of damage initiation, t=1.005 s, (**c**) the instant with the maximum damage, t=1.7 s.

**Figure 12 sensors-23-07926-f012:**
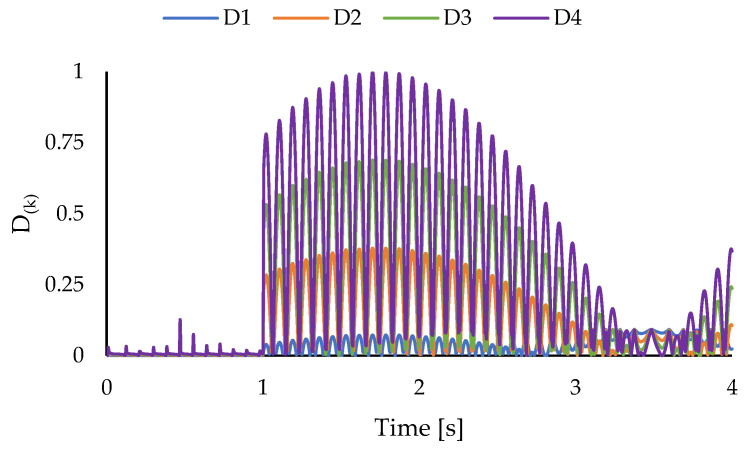
Layer-wise damage indices for cantilevered plate under harmonic torsion load.

**Figure 13 sensors-23-07926-f013:**
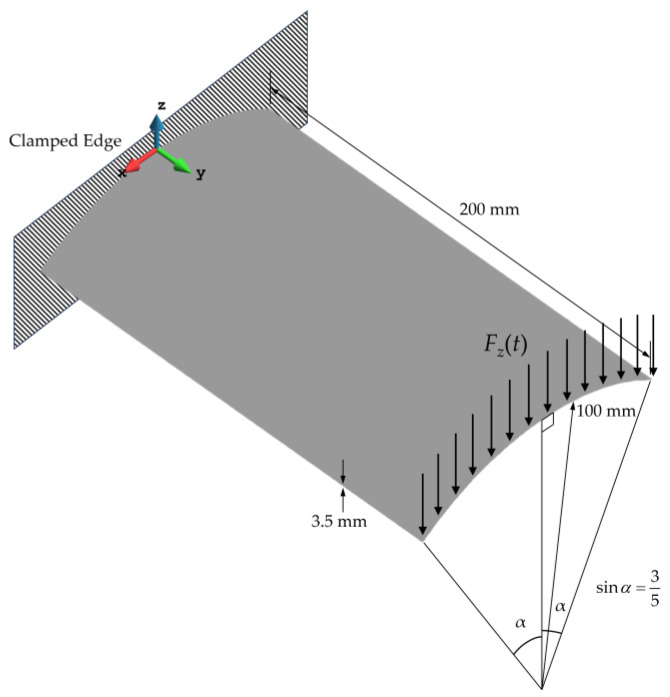
Cantilevered curved laminated shell subject to random bending load.

**Figure 14 sensors-23-07926-f014:**
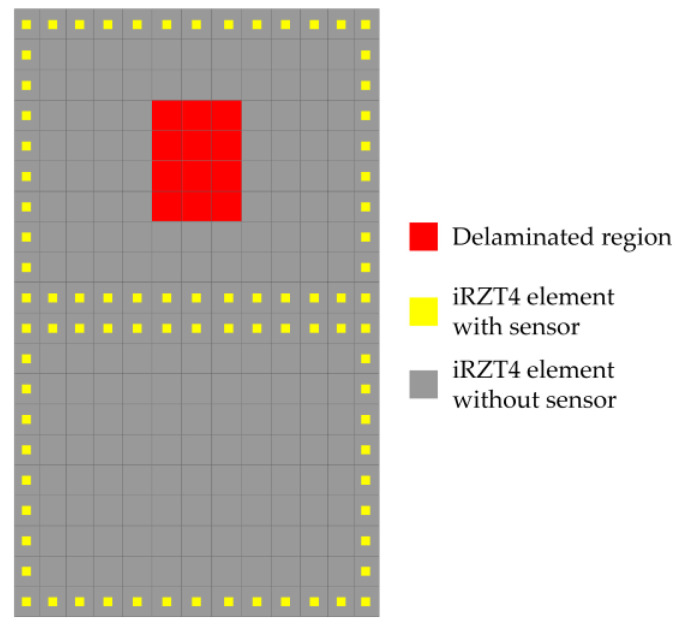
Sensor placement scheme and in-plane location of the delaminated region provided in (x–y) plane.

**Figure 15 sensors-23-07926-f015:**
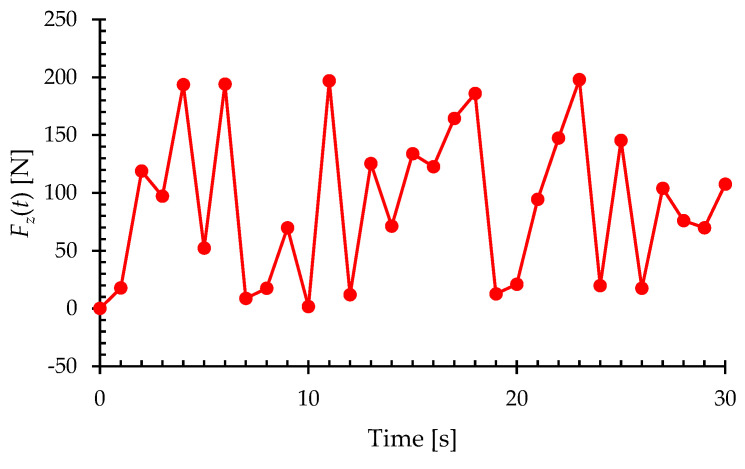
Random variation in the bending load with respect to the time of the analysis.

**Figure 16 sensors-23-07926-f016:**
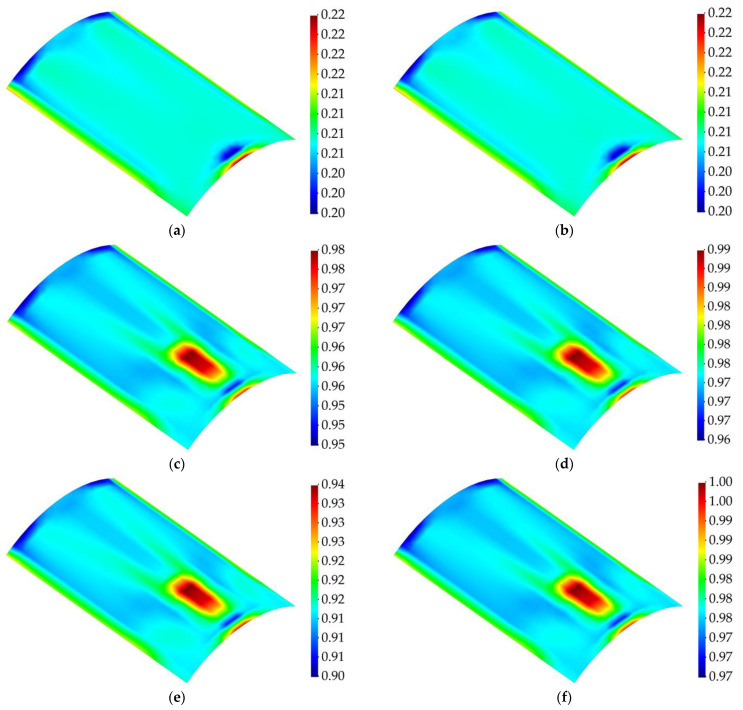
Membrane damage index for cantilevered curved shell subject to random bending load; (**a**) t=4 s, (**b**) t=5 s (Last time step prior to damage initiation), (**c**) t=7 s, (**d**) t=11 s, (**e**) t=18 s, and (**f**) t=23 s.

**Figure 17 sensors-23-07926-f017:**
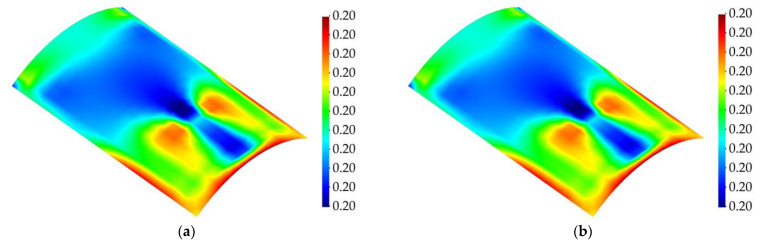
Twisting damage index for cantilevered curved shell subject to random bending load; (**a**) t=4 s, (**b**) t=5 s (Last time step prior to damage initiation), (**c**) t=7 s, (**d**) t=11 s, (**e**) t=18 s, and (**f**) t=23 s.

**Figure 18 sensors-23-07926-f018:**
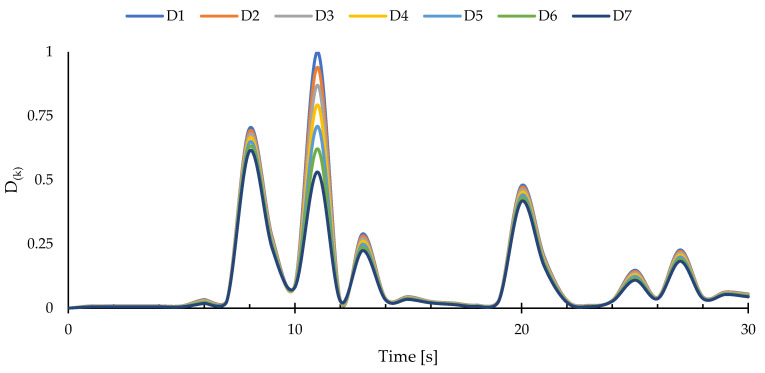
Layer-wise damage indices for cantilevered curved shell subject to random bending load.

**Table 1 sensors-23-07926-t001:** Material properties of an individual unidirectional carbon–epoxy ply.

Young’s Modulus [GPa]	Poisson’s Ratio	Shear Modulus [GPa]
$E_{1}=133.9$	$\nu_{12}=\nu_{13}=0.32$	$G_{12}=G_{13}=4.8$
$E_{2}=E_{3}=11.5$	$\nu_{23}=0.37$	$G_{23}=4.2$

## Data Availability

The data will be made available upon request.

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
