# Peer review of "Delamination Detection and Localization in Vibrating Composite Plates and Shells Using the Inverse Finite Element Method"

_sensors, 2023, doi:10.3390/s23187926_

Round 1

Reviewer 1 Report

Abstract  (line 17). Von Mises stress is used. When dealing with delamination explained in physical terms this is quite incorrect (what drives delamination is peel stress and out of plane shear). Please remark you are rather using it as an in plane stress index.  Please remark also that you are working with a specific category of composite: laminates (not 2d or 3d woven). 

Introduction (after line 42). Please remark that delamination may be caused by the stacking sequence itself (please cite doi 10.2174/1874155X01812010151) . 

Theoretical Framework (line 182). Not clear what psi is (zig-zag rotation). Please add a figure to explain it. I know you refer to [33], but the understanding of this kinematic parameter is quite fundamental and it should be clarified to the reader.

Damage detection toolbox based on iFEM-RZT (line 287) . Again on the Von Mises. Please refer to the comment of line 17. Will this index work for a thick chunky structure instead of a plate?

Numerical Example(after line 334). Please remark that this method is good for crack initiation on not for delamination propagation.

Line 345 and 346: Refer to Mode I and Mode II (first flap, first torsion). Furthermore, please show how the delamination has been modelled in the FEM (simply lack of layer? a cohesive layer instead?)

Conclusions (please discuss the limitation of this approach). Is it valid for general composite structures?

Reviewer 2 Report

In this paper, a delamination detection approach based on equivalent von Mises strains has been demonstrated for vibrating composite plates. The governing relations of the inverse finite element method have been recast according to the refined zigzag theory. Using the in-situ strain measurements obtained from the surface and through the thickness of the composite shell, the inverse analysis is performed, and the strain field of the composite shell is reconstructed. The implementation of the proposed methodology has been demonstrated for two numerical case studies associated with the harmonic and random vibrations of composite shells. It can be accepted after revision.

1.      The conclusion part is too long, and needs to be condensed to reflect some quantitative conclusions.

2.      The introduction part is too long, and needs to be refined to reflect innovation.

3.      How can this technology be applied to online damage detection?

4.      It is recommended to conduct some experiments to verify the reliability of the conclusions.

Reviewer 3 Report

In the paper, authors have e proposes a delamination detection approach based on equivalent von Mises strains has been demonstrated for vibrating composite plates. In my opinion this paper needs minor revision. Detailed comments and questions are provided below:

Comment (1):    The introduction section should be improved by discussing the existing methods of delamination detection.

Comment (2):  The model validity should be shown by comparing the results with currently published articles.

Comment (3):  The limitation of the proposed method should be discussed.

Comment (4):  The Conclusion section should be improved by summarizing contributions of the paper.

Round 2

Reviewer 2 Report

It can be accepted in present form.